# Knockouts of *TNFRSF1A* and *TNFRSF1B* Genes in K562 Cell Line Lead to Diverse Long-Lasting Responses to TNF-α

**DOI:** 10.3390/ijms242417169

**Published:** 2023-12-06

**Authors:** Olga Perik-Zavodskaia, Saleh Alrhmoun, Roman Perik-Zavodskii, Julia Zhukova, Julia Lopatnikova, Marina Volynets, Alina Alshevskaya, Sergey Sennikov

**Affiliations:** 1Laboratory of Molecular Immunology, Research Institute of Fundamental and Clinical Immunology, 630099 Novosibirsk, Russiasaleh.alrhmoun1@gmail.com (S.A.); zhukova1982@rambler.ru (J.Z.); lopatnikova18@yandex.ru (J.L.); mrsmarinavolynets@gmail.com (M.V.); 2Laboratory of Immune Engineering, Federal State Autonomous Educational Institution of Higher Education I.M. Sechenov First Moscow State Medical University of the Ministry of Health of the Russian Federation (Sechenov University), 119991 Moscow, Russia

**Keywords:** TNFR1, TNFR2, *TNFRSF1A*, *TNFRSF1B*, knockout, cell line model, K562, gene expression

## Abstract

This research delves into the intricate landscape of tumor necrosis factor-alpha (TNF-α) signaling, a multi-functional cytokine known for its diverse cellular effects. Specifically, we investigate the roles of two TNF receptors, TNFR1 and TNFR2, in mediating TNF-α-induced transcriptional responses. Using human K562 cell lines with TNFR1 and TNFR2 knockouts, we explore changes in gene expression patterns following TNF-α stimulation. Our findings reveal distinct transcriptional profiles in TNFR1 and TNFR2 knockout cells, shedding light on the unique contributions of these receptors to TNF-α signaling. Notably, several key pathways associated with inflammation, apoptosis, and cell proliferation exhibit altered regulation in the absence of TNFR1 or TNFR2. This study provides valuable insights into the intricate mechanisms governing TNF-α signaling and its diverse cellular effects, with potential implications for targeted therapeutic strategies.

## 1. Introduction

Tumor necrosis factor alpha (TNF-α) is a multi-functional pro-inflammatory cytokine that plays a key role in physiological and pathological processes [1]. This cytokine not only controls inflammatory processes but also regulates various cellular processes, including proliferation and cell differentiation, as well as apoptosis and necroptosis [2,3]. The wide range of biological effects of TNF-α is possible due to the presence of two types of typical competing receptors: TNFR1 and TNFR2 (encoded by the *TNFRSF1A* and *TNFRSF1B* genes, respectively), which activate different intracellular signaling pathways [4].

Despite many hypotheses, there is still no sophisticated understanding of how TNF-α mediates its diverse biological effects [2,5]. One of the key hypotheses is the assumption of differential interactions between different forms of cytokines and receptor complexes, including soluble and membrane-bound forms [6]. Another hypothesis suggests that the type of regulation and efficiency of interaction may be possible by changing the ratio between ligands and receptors. The study of cytokine levels on the realized effects has been proven both in experimental models and in clinical studies [7], while the number and ratio of different types of receptors are being actively investigated [8,9].

An experimental model on human cell lines with a knockout of one type of receptor allows us to evaluate the contribution of each type of receptor to the activation of certain signaling pathways and the possibility of isolated interaction of cytokines with each receptor. The triggering of signaling pathways in the body is a complex and multi-step process intricately dependent on specific activation mechanisms, including the characterization of the interaction between various forms of cytokines and receptor complexes [10,11], identification of different types of receptors [12], and the ligand–receptor ratio [7].

A recent murine TNFR1 and TNFR2 knockout study on iTreg [13] showed that TNFR2 deficiency “hampered iTreg differentiation, proliferation, and function”, while TNFR1 deficiency “decreased the differentiation of inflammatory T cells such as Th1 and Th17 cells”, thus highlighting the possible differences of the TNF signaling.

Thus, the purpose of this study was to investigate the impact of TNF-α stimulation on the transcriptomic activity of human K562 cells with gene knockouts of TNF receptor types 1 and 2.

## 2. Results

### 2.1. Transcriptomic Analysis of TNFR1 and TNFR2 Signaling Pathways in the K562 Cell Line

In this study, we conducted a comprehensive transcriptomic analysis of the K562 cell line, focusing on gene expression. The gene expression data, presented in descending order of copy numbers detected, included: *TFRC*, *PSMB7*, *B2M*, *C1QBP*, *IFITM1*, *ILF3*, *PSMD7*, *CTNNB1*, *PSMC2*, *C14orf166*, *MCL1*, *CD81*, *MAPK1*, *CTSC*, *CD164*, *PSMB5*, *FN1*, *UBE2L3*, *CD46*, *GP1BB*, *STAT5A*, *FKBP5*, *ITGB1*, *MIF*, *STAT3*, *TGFBR1*, *NFKBIA*, *MAP4K4*, *ATG5*, *BCAP31*, *CD99*, *NFATC3*, *PSMB8*, *CD3D*, *STAT5B*, *RAF1*, *CD59*, *CFH*, *ARHGDIB*, *CHUK*, *CD44*, *PTK2*, *CCND3*, *CD53*, *MAPKAPK2*, *BCL10*, *TGFB1*, *CTSS*, *CD58*, *FCGR2A*, *CASP2*, *TAL1*, *BAX*, *ITGA5*, *IL2RG*, *GPI*, *SMAD5*, *FYN*, *SKI*, *TMEM173*, *ABL1*, *IFNGR1*, *IL13RA1*, *TRAF4*, *STAT2*, *MAPK14*, *CD97*, *MYD88*, *IKBKAP*, *LCP2*, *TRAF5*, *ICAM3*, *TYK2*, *PSMB9*, *AHR*, *IRAK1*, *XBP1*, *TBK1*, *FCGRT*, *JAK1*, *BST2*, *CD45R0*, *KIT*, *ATG16L1*, *SYK*, *NFIL3*, *EGR1*, *SMAD3*, *JAK2*, *SLC2A1*, *IL6ST*, *LTBR*, *TRAF2*, *IFNAR2*, *CFD*, *TAPBP*, *RELA*, *IL8*, *MALT1*, *ITGAE*, *ADA*, *CD9*, *PLAU*, *BTK*, *ICAM4*, *ATG10*, *LEF1*, *NFATC2*, *PTPN2*, *PDCD2*, *BCL2L11*, *ICOSLG*, *CD3EAP*, *LGALS3*, *LITAF*, *CASP3*, *CD276*, *CRADD*, *IKBKB*, *PML*, *TNFAIP3*, *RUNX1*, *TCF4*, *CD24*, *TNFRSF1B*, *CUL9*, *IGF2R*, *ARG2*, *KLRC3*, *TNFRSF8*, *STAT1*, *ITGA2B*, *TRAF3*, *MUC1*, *IKZF2*, *IKZF1*, *TICAM1*, *CISH*, *ICAM1*, *IRF1*, *TGFBR2*, *IRAK4*, *ATG7*, *NFKB2*, *ICAM2*, *TAP1*, *HRAS*, *STAT6*, *TAP2*, *IKBKG*, *CEBPB*, *PTPN22*, *CCBP2*, *CASP8*, *TCF7*, *TRAF6*, *IRF3*, *MAP4K2*, *IKBKE*, *EDNRB*, *CDKN1A*, *NFATC1*, *TOLLIP*, *ATG12*, *IL18R1*, *NFKB1*, *ZBTB16*, and *NOTCH2*.

Among the detected genes, NFKB2, RELB, CASP8, NFKB1, CEBPB, NFKBIA, TNFRSF1B, TRAF5, MAPK1, IKBKB, RELA, TRAF3, IKBKG, MAPK14, IRF1, CHUK, TRAF2, and CASP3 were identified as components of the TNF-α signaling pathway from TNFR1 and TNFR2.

We subsequently performed a gene enrichment analysis and constructed a small gene network of the TNF-α signaling pathway for TNFR1 and TNFR2 in K562 cell line in Cytoscape incorporating the above genes, enriched in “KEGG TNF signaling pathway” term, and imputing the TNFRSF1A gene using the STRING database (Figure 1).

An analysis of the signaling pathways, starting from the TNF-α receptors revealed the potential activation of the NF-κB signaling pathway via both the canonical (*RELA* and *NFKB1*) [14] and the alternative pathways (*RELB* and *NFKB2*) [15], which contribute to cell survival and the synthesis of pro-inflammatory cytokines [16]. As well, the data suggested possible activation of apoptosis through TNFR1 (involving *CASP8* and *CASP3*) [17].

### 2.2. Transcriptomic Analysis of K562 with TNFR1 and TNFR2 Knockouts and TNF-α Addition

The exposure of K562 cells to TNF-α itself resulted in a notable decreased expression of the *FCGR2A*, *CD3D*, and *CDKN1A* genes after 72 h (Figure 2a). Gene set enrichment analysis of these down-regulated genes showed an enrichment in cell cycle transition-related Gene Ontology “Biological Process” terms (Figure 2b).

The knockout of TNFR1 (KIA) did not lead to significant changes in gene expression, except for an increase in CD45RO gene expression (Figure 3a). However, the knockout of TNFR2 (KIB) revealed distinct alterations; after 72 h, the expression of *CD24*, *FCGR2A*, *FN1*, *IL8*, *TAP1*, *LGALS3*, *ICOSLG*, *EGR1*, *PSMB8*, and *PSMB9* increased, while the expression of *MUC1*, *NOTCH2*, *CD45RO*, *CFH*, and *GP1BB* decreased (Figure 3b). After the addition of TNF-α to the TNFR1 knockout K562 cell culture, a similar pattern of gene expression was observed as with the TNFR2 knockout, but without the addition of TNF-α—the expression of the *FN1* and *IL8* genes increased, and the expression of the *NOTCH2*, *CD45RO*, and *CD3D* genes decreased (Figure 4a). A gene set enrichment analysis of these up-regulated genes showed an enrichment in cell proliferation-related Gene Ontology “Biological Process” terms (Figure 4b).

Remarkably, the knockout of TNFR2 with the addition of TNF-α after 72 h resulted in a unique pattern of gene expression changes. Following this intervention, the expression of the genes *CD45RO*, *MUC1*, *CD24*, *CD44*, *GP1BB*, *TAPBP*, *KIT*, *CGH*, *CFD*, and *LCP2* increased, while the expression of *FCGR2A* and *EGR1* decreased (Figure 4c). A gene set enrichment analysis of these down-regulated genes showed an enrichment in negative regulation of apoptosis-related Gene Ontology “Biological Process” terms (Figure 4d).

To summarize these findings, we created a heatmap illustrating the differential gene expression data (Figure 5).

## 3. Discussion

In this study, we conducted an investigation into gene expression patterns following TNFR1 and TNFR2 knockouts within the K562 cell culture.

A reconstruction of a small gene network of signaling from TNFR1 and TNFR2 suggested the presence of multi-directional responses to TNF-α—proliferative, anti-proliferative—and associated with the synthesis of proinflammatory cytokines.

Upon the addition of TNF-α to K562 cells without knockouts, the expression of the *CDKN1A* gene, an inhibitor of cyclin/CDK complexes, decreased, suggesting a proliferative response due to a free cell cycle progression in reaction to TNF-α. 

The knockout of TNFR2 in K562 produced a similar transcriptomic effect as the knockout of TNFR1 with the addition of TNF-α, in which signaling occurs through TNFR2. This suggests that knocking out TNFR2 potentially locks it into the activated position for cultured K562 cells. 

All detected differentially expressed genes were not directly affiliated with the signaling pathways from TNFR1 and TNFR2 and, apparently, reflect long-term changes in the transcriptome in response to TNF-α; the TNFR1 knockout with the addition of TNF-α and the TNFR2 knockout led to a pro-inflammatory response type with an increased expression of the *FN1* and *IL8* genes [16,18,19], while the knockout of TNFR2 with the addition of TNF-α led to an increased expression of the *KIT* proto-oncogene, suggesting a proliferative response [20].

Overall, the knockouts of TNFR1 and TNFR2 induced noticeable alterations in the K562 cell transcriptome, and the addition of TNF-α further modulated such changes, ranging from pro-proliferative to pro-inflammatory.

## 4. Materials and Methods

### 4.1. Cell Culture

The K562 cell line was provided by the Federal State Budgetary Institution of Science, Institute of Cytology of the Russian Academy of Sciences (FGBI INC RAS, St. Petersburg, Russia). The ampoules with cells were thawed in a water bath at a temperature of +37 °C for 1–2 min. Then, the cells were washed once and cultured in RPMI-1640 medium containing 10% fetal bovine serum (FBS) (HyClone, Logan, UT, USA), 2 mM L-glutamine (BioloT LLC, Saint-Petersburg, Russia), 5 × 10^−4^ M2-mercaptoethanol (Sigma-Aldrich, St. Louis, MO, USA), 80 µg/mL gentamicin (KRKA, Novo mesto, Slovenia), 10 mM HEPES buffer (Sigma-Aldrich, USA), and 100 µg/mL benzylpenicillin (JSC Biosintez, Penza, Russia) in an incubator in a humid atmosphere at 37 °C and a CO_2_ concentration of 5%. Cells were collected by washing away the cells with a liquid stream using a pipette. After washing, the cells were collected and centrifuged for 10 min at 1500 rpm, followed by re-suspension in complete medium and transferred into vials (TPP, Trasadingen, Switzerland) at the optimal density for each cell line.

### 4.2. TNFRSF1A and TNFRSF1B Gene Knockouts

For each target gene, two single-guide RNAs (sgRNAs) were designed using bioinformatics tools, including Crispor (available at crispor.org, accessed on 23 October 2023), Benchling (benchling.com, accessed on 23 October 2023), and CHOPCHOP (chopchop.cbu.uib.no, accessed on 23 October 2023). These sgRNAs were strategically designed to specifically target the genomic loci of *TNFRSF1A* and *TNFRSF1B*.

To create the sgRNA expression vectors, oligo-nucleotides encoding these sgRNAs were synthesized and further processed to generate the necessary compatible sticky ends for ligation. Each sgRNA-coding oligonucleotide was designed to match the respective target gene, ensuring precise and selective gene knockout.

The next step involved ligating these sgRNA-coding oligo-nucleotides into a suitable and well-characterized expression vector, pSpCas9(BB)-2A-GFP (Addgene, #48138, Watertown, MA, USA). The ligation process was carried out using restriction enzymes and their corresponding recognition sites within the plasmid. Following ligation, the recombinant plasmids were introduced into *E. coli* NEB stable cells, where the cells were transformed with the ligated constructs. This process enabled the creation of a library of colonies that contained the assembled plasmids targeting both *TNFRSF1A* and *TNFRSF1B* genes.

To confirm the successful assembly and integrity of the plasmids, colonies were subjected to thorough screening. PCR analysis was employed to confirm the presence of the sgRNAs within the plasmids. Additionally, the structure and fidelity of the final plasmids were confirmed through Sanger sequencing, ensuring that the knockout strategy was both accurate and specific. The effectiveness and stability of knockouts were confirmed using flow cytometry, and the TNFR1 and TNFR2 were absent on cell membranes during subsequent passages. The expression of TNFR1 and TNFR2 was assessed by using flow cytometry (AttuneNxT cytofluorimeter (ThermoFisher, Waltham, MA, USA)) using monoclonal antibodies: anti-human CD120a-PE and anti-human CD120b-APC (R&D Systems, Minneapolis, MN, USA). Data processing and calculation of fluorescence intensity indicators were carried out using Attune™NxT Software version 3.2.1 (ThermoFisher, Waltham, MA, USA) (Figure 1).

### 4.3. TNF-α Co-Culturing

Cell lines were cultured either in the presence of recombinant TNF (R&D Systems, USA) at a concentration of 5 ng/mL or simply without anything for 72 h. 

### 4.4. Total RNA Extraction

We isolated total RNA from 250,000 cells with the Total RNA Purification Plus Kit (Norgen Biotek, Thorold, ON, Canada). We then measured the concentration and assessed the quality of the total RNA in each sample on a NanoDrop 2000c (Thermo Fisher Scientific, USA). The total RNA samples were frozen at −80 °C until the gene expression analysis.

### 4.5. NanoString RNA Profiling

We performed gene expression profiling with the help of the Nanostring nCounter SPRINT Profiler analytical system using 100 ng of total RNA from each sample. We used nCounter Human Immunology v2 panel to analyze the total RNA samples. nCounter Human Immunology v2 panel consisted of 579 immune- and inflammation-associated genes, 15 housekeeping genes and eight negative and six positive controls. The samples (*n* = 3) were subjected to a 20 h hybridization reaction at 65 °C, where 5–14 μL of total RNA was combined with 3 μL of nCounter Reporter probes, 0–7 μL of DEPC-treated water, 11 μL of hybridization buffer, and 5 μL of nCounter capture probes (total reaction volume = 33 μL). After the hybridization of the probes to targets of interest in the samples, the number of target molecules was determined on the nCounter digital analyzer. We performed normalization and QC in nSolver version 4.0 using the positive controls and the 15 housekeeping genes included in the panel. We then performed background thresholding on the normalized data to remove non-expressing genes. The background level was determined as the mean of the POS_E controls. We then log2-transformed the data.

### 4.6. TNFR1 and TNFR2 Signaling Gene Network Reconstruction for K562

We used Cytoscape version 17.0 [21] for the TNF signaling gene network reconstruction. We imported a new network from the STRING database [22] with a query containing every gene with detected expression. We then performed STRING functional enrichment of the network in KEGG pathway terms and created a sub-network of genes enriched in the “KEGG TNF signaling pathway” term. We used 0.95 protein–protein interaction confidence for the edge creation.

### 4.7. Differential Gene Expression Testing

We performed differential gene expression using multiple *t*-tests (with Q < 0.05) in GraphPad Prism version 9.4. The Volcano plots were created in GraphPad Prism 9.4. The heatmap of the differentially expressed genes across all conditions was created using bioinfokit (https://github.com/reneshbedre/bioinfokit, accessed on 23 October 2023). We then performed gene ontology biological process enrichment analysis (GOEA) of the detected genes via GSEApy version 1.0.0 [23].

## Data Availability

The datasets generated and analyzed during the current research are accessible from the corresponding author upon an email request.

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
