# Peer review of "Knockouts of TNFRSF1A and TNFRSF1B Genes in K562 Cell Line Lead to Diverse Long-Lasting Responses to TNF-α"

_ijms, 2023, doi:10.3390/ijms242417169_

Round 1
Reviewer 1 Report
Comments and Suggestions for Authors
Comments
This is a well-written manuscript titled “Knockouts of TNFRSF1A and TNFRSF1B Genes in K562 Cell 2 Line Lead to Diverse Long-Lasting Responses to TNF-α”. In this study, the authors demonstrated the roles of two TNF receptors, TNFR1 and TNFR2, in mediating TNF-α-induced transcriptional responses. The author has used human K562 cell lines with TNFR1 and TNFR2 knockouts and showed the changes in gene expression patterns following TNF-α stimulation. Overall studies revealed that distinct transcriptional profiles in TNFR1 and TNFR2 knockout cells, raise the possibility of unique contributions of these receptors to TNF-α signaling. Pathways associated with inflammation, apoptosis, and cell proliferation exhibit altered regulation in the absence of TNFR1 or TNFR2. This study overall demonstrated valuable insights into the intricate mechanisms governing TNF-α signaling and its 20 diverse cellular effects, with potential implications for targeted therapeutic strategies. The figures/ tables/ schemes are appropriate and align with the text and in present form. This manuscript also identified gaps in knowledge and will be of good interest to the scientific community. The manuscript can be accepted in its present form with minor revisions.
1. The figures/ tables/schemes are self-explanatory and nicely illustrated in present review. It will be helpful for the reader if the author validates a few significant transcriptional changes of K562 with TNFR1 and TNFR2 knockouts and TNF-α addition by RT-QPCR.
2. It will be helpful for the reader if the author shows some significant changes in K562 with TNFR1 and TNFR2 knockouts and TNF-α at the protein level by immunoblot.
Comments on the Quality of English Language
The manuscript is well written and needs minor editing like spelling mistakes.
Author Response
Dear Reviewer,
Thank you for your constructive feedback on our manuscript.
Regarding the points you suggested:
>1. The figures/ tables/schemes are self-explanatory and nicely illustrated in present review. It will be helpful for the reader if the author validates a few significant transcriptional changes of K562 with TNFR1 and TNFR2 knockouts and TNF-α addition by RT-QPCR.
We appreciate your suggestion to validate our NanoString results using RT-qPCR. While we acknowledge the value of corroborating findings through different methodologies, we chose not to perform RT-qPCR in this study due to the established high correlation between NanoString and RT-qPCR results, as demonstrated by previous research (https://bmcbiotechnol.biomedcentral.com/articles/10.1186/1472-6750-11-46), which reported a correlation coefficient of 0.8. Moreover, our NanoString data analysis included stringent noise removal criteria, enhancing the reliability of our transcriptional changes. We believe that these measures ensure the robustness of our findings without the need for additional validation.
>2. It will be helpful for the reader if the author shows some significant changes in K562 with TNFR1 and TNFR2 knockouts and TNF-α at the protein level by immunoblot.
We acknowledge the importance of investigating protein-level changes. Unfortunately, we currently face constraints in performing immunoblot studies due to the unavailability of specific reagents. However, we are committed to addressing this aspect in our future studies through a planned proteomic experiment. In the interim, we have supplemented our findings with flow cytometry data, providing evidence of TNFR1 and TNFR2 knockouts. We believe these data contribute valuable insights into the functional consequences of the genetic modifications under investigation.
Once again, we appreciate your thoughtful comments and will ensure that our future work addresses the aspects you have highlighted.
Reviewer 2 Report
Comments and Suggestions for Authors
In this study, the authors tried to decipher the transcriptomic landscape of a human derived cancerous cell-line (K562) knocked-out for TNFR1 or TNFR2 and tried to distinguish different patterns of gene dysregulation upon TNF stimulation of these two isolated receptors. Although the concept is interesting, I am afraid that the presentation of the results and the results in itself is not sufficient in its state for publication.
Major points.
The first part of the main manuscript is empty of information as is Figure 1 which just takes a result from STRINGR and puts it as a Figure. The authors deliver a very big list of genes which is not interesting for the reader in this format.
It is absolutely not clear from which cells in which condition the cited genes are up or downregulated. The manuscript should be completely revised to have sentences with a hypothesis, way of answering it, and answer.
The figure 1 is empty of relevant information.
The authors should provide gene set enrichment analysis to decipher which pathway is impacted by the knock-out.
Authors claims that they verified by Sanger and Flow Cytometry the KO. Please show the data in supplementary as this information is critical.
Authors should discuss more the mice studies with TNFR1 or TNFR2 KO.
Author Response
Dear Reviewer,
Thank you for your thorough review of our manuscript.
Regarding the points you suggested:
>The first part of the main manuscript is empty of information as is Figure 1 which just takes a result from STRINGR and puts it as a Figure!
We appreciate your concern about the clarity of the first part of the main manuscript and Figure 1. As outlined in the Materials and Methods section, we conducted enrichment analysis and selectively retained genes associated with the TNF signaling pathway. To enhance clarity, we will provide additional details in the Results section to better contextualize the information presented in Figure 1.
>It is absolutely not clear from which cells in which condition the cited genes are up- or down-regulated.
We acknowledge your feedback regarding the lack of clarity on the cell type and conditions in which genes are up- or down-regulated. In response, we have restructured Figure 2 into three separate figures, explicitly detailing the relevant cell types and conditions to address this concern.
>The manuscript should be completely revised to have sentences with a hypothesis, way of answering it, and answer.
Your suggestion to revise the manuscript for a more structured presentation of hypotheses, methods, and results is well taken. We have incorporated additional sentences following this structure throughout the manuscript to enhance the clarity of our research narrative.
>The figure 1 is empty of relevant information.
We understand your comment on Figure 1, and we would like to clarify that its purpose is to visually confirm that K562 cells exhibit differential responses to TNF-alpha. We will provide additional context and explanations in the manuscript to underscore the significance of Figure 1.
>The authors should provide gene set enrichment analysis to decipher which pathway is impacted by the knock-out.
To address your recommendation, we have integrated GSEA into the differential gene expression (DGE) analysis results whenever applicable. This addition provides a more comprehensive understanding of the pathways impacted by the knockouts.
>Authors claims that they verified by Sanger and Flow Cytometry the KO. Please show the data in supplementary as this information is critical.
We have included flow cytometry data confirming the knockouts in the supplementary material. Please note that we did not employ Sanger sequencing for quality control, as surface protein levels were deemed more reliable indicators of knockout functionality.
>Authors should discuss more the mice studies with TNFR1 or TNFR2 KO.
Following your suggestion, we have incorporated a more in-depth discussion of the mice studies with TNFR1 or TNFR2 knockout in the introduction section, providing a more thorough overview of our experimental approach and rationale.
We sincerely appreciate your valuable feedback, and believe that the revisions made adequately address the concerns raised in your review.